# Recent Approaches to Design and Analysis of Electrical Impedance Systems for Single Cells Using Machine Learning

**DOI:** 10.3390/s23135990

**Published:** 2023-06-28

**Authors:** Caroline Ferguson, Yu Zhang, Cristiano Palego, Xuanhong Cheng

**Affiliations:** 1Department of Bioengineering, Lehigh University, Bethlehem, PA 18015, USA; cal618@lehigh.edu (C.F.); yuzi20@lehigh.edu (Y.Z.); 2Department of Computer Science and Electronic Engineering, Bangor University, Bangor LL57 2DG, UK; c.palego@bangor.ac.uk; 3Department of Materials Science and Engineering, Lehigh University, Bethlehem, PA 18015, USA

**Keywords:** machine learning, electrical sensing, single-cell analysis, impedance cytometry, impedance spectroscopy

## Abstract

Individual cells have many unique properties that can be quantified to develop a holistic understanding of a population. This can include understanding population characteristics, identifying subpopulations, or elucidating outlier characteristics that may be indicators of disease. Electrical impedance measurements are rapid and label-free for the monitoring of single cells and generate large datasets of many cells at single or multiple frequencies. To increase the accuracy and sensitivity of measurements and define the relationships between impedance and biological features, many electrical measurement systems have incorporated machine learning (ML) paradigms for control and analysis. Considering the difficulty capturing complex relationships using traditional modelling and statistical methods due to population heterogeneity, ML offers an exciting approach to the systemic collection and analysis of electrical properties in a data-driven way. In this work, we discuss incorporation of ML to improve the field of electrical single cell analysis by addressing the design challenges to manipulate single cells and sophisticated analysis of electrical properties that distinguish cellular changes. Looking forward, we emphasize the opportunity to build on integrated systems to address common challenges in data quality and generalizability to save time and resources at every step in electrical measurement of single cells.

## 1. Introduction

### 1.1. Motivation to Measure Single Cells

The uniqueness of gene expression and phenotype is inherent in any biological system and generates the variation of function necessary to maintain homeostasis in our cells and bodies. Recent work in the field of healthcare has sought to address the need to personalize medicine and design diagnostics that are flexible and sensitive to variations between patients and between individual cells making up a single system, especially in the context of resource-limited areas [1,2]. For example, there is a need to identify circulating tumor cells (CTCs) from the other cells that make up the composition of a blood sample to predict cancer prognosis [3]. While size can act as a preliminary method for isolating certain components of blood, more complex methods are needed to tease apart the identity and origin of CTCs from cells with similar size [4]. Even within a single organ, the population of cells is composed of individuals, each with unique genetic and physiological properties. For this reason, the measurement of single cells and analysis of population heterogeneity has become a focus of modern diagnostics research. Beyond the expansion of technology into resource-limited areas, the movement towards personalized healthcare has been essential in identifying the benefits of single-cell analysis. A review from Tavakoli et al. describes the way recent advances in microfluidics have empowered the study of single-cell applications in the context of cancer understanding, diagnosis, therapy, highlighting the necessity of individual measurements [5]. Similar efforts studying other diseases have used microfluidics to isolate and genetically analyze single cells while reducing the equipment and footprint necessary [6,7].

Cellular heterogeneity comes from variations in genetics or expression that can be caused by random mutations or as a response to environmental factors. Heterogeneity poses many challenges for both measurement processes and the design of analytical systems. A single cell measurement system must have the sensitivity to capture specific small-scale changes, sufficient data features, and a sample size to detect these nuances. Similarly, an analytical system needs the capability to handle a large volume of data and often requires more sophisticated approaches than purely statistical analysis. When studying a population of single cells, data tends to be more dispersed, rather than the cleanly defined data belonging to less heterogeneous systems. In the study of cellular populations, it is important to have the ability to identify not just important features and trends, but also determine standout or outlier cells in a population that may not be representative of the whole [8,9]. When looking at consistent and integrative methods to generate such data rapidly and with minimal resources, a natural choice is evaluation of the cellular electrical properties. 

### 1.2. High-Throughput Electrical Measurement

For the evaluation of single cells making up a larger population, the necessary number of measurements is limited by the techniques used to manipulate the cells physically and measure their properties. Electrical impedance measurements using microfluidic channels has become a popular mechanism for single-cell handling because of the ability to design precise control of the cell measurement location and due the rapid nature of the electrical signal acquisition [10,11]. These systems also have the potential to add physical, chemical and immunological cell property measurements using optical systems, and to probe mechanical, inertial and adhesive characteristics through microfluidic designs for a rapid and multi-faceted approach to characterization [12]. Although methods exist to look at individual cell properties using optical and genetic profiling techniques, these techniques are less diagnostically accessible than electrical cell profiling. Electrical characterization has the benefit of not requiring label molecules, rapid sample preparation and measurement, and low-profile devices that are easily translatable to point-of-care purposes. Although the electrical measurements tend to give less specific information, the variety of available experimental parameters and variables is ideal for the incorporation of machine learning algorithm adoption.

The inclusion of multiple types of electronic sensor designs generates highly tunable systems which can maximize the multi-frequency information obtained from each cell during its travel in the channel [13]. Additionally, microfluidic systems have been used to isolate chambers for simultaneous measurements of multiple samples at a time. Lopez et al., reported a multi-cell sensor capable of measuring constant current stimulation, constant voltage stimulation, and impedance spectroscopy on roughly 16,500 input electrodes with integrated analysis [14]. The combination of electrical measurement and microfluidics is paramount for the development of lab-on-a-chip devices that can incorporate the handling, measurement, and analysis of samples on a small footprint. Such integrative devices have gained popularity as accessibility has become a goal in the healthcare field because they have the potential to function in areas lacking resources in infrastructure, personnel, or consumables.

### 1.3. Machine Learning Applications in Studying Complex Variable Relationships

Because cell individuality can influence a variety of cell properties and processes as summarized in Figure 1, analysis can require spatial, temporal, or multimodal data. The data required to capture deep understanding of a population lends itself to machine learning as an analytical tool, especially in conditions with many input modalities or when comparing highly overlapping population changes. Machine learning is the study of learning processes and application of computer-based modeling to fit and predict trends in large datasets [15]. Exemplifying the capability for machine learning to address the challenges associated with single cells, Chien et al., showed that single cells with highly overlapping electrical opacity can be distinguished visually using clustering and population distribution even with no significant difference by statistical analysis, demonstrating the capabilities of clustering algorithm or principal component analysis to find new relationships in single-cell data [16]. Traditional machine learning using feature selection, classification, or a combination thereof can give information on both the most important features to identify changes and subsequently inform future iterations of device designs. 

Beyond classical machine learning algorithms, multilayer artificial neural networks can add layered decision-making processes to model more complex relationships, similar to the way human neurons process information, using variable feature information and context to generate understanding. Deep learning has expanded the capabilities of the machine learning field to enable more adaptable algorithms using larger and more varied sets of data. Deep learning incorporation is key to developing precision and individualized medicine in a clinical setting as shown in previous work using biological measurements or imaging data to predict disease state of an individual. Deep learning has also expanded past the scientific fields to incorporation in our daily life in audio processing, facial recognition, and data retrieval by search engines [17]. The benefit of its application in comparison to traditional statistical methods is the ability to parse complex relationships between many variables in applications like predicting human behavior and determining how the combination of these variables contributes to an overall classification or outcome [18,19]. The trained model can often be used to generate optimized variable values or improve visualizations to show distinctions in an otherwise convoluted dataset. For this reason, neural networks are commonly applied to the study of single-cell characteristics making up larger, often heavily overlapping populations [20,21,22]. As cellular measurements can include larger amounts of data either in fluorescence at multiple wavelengths, optical monitoring over periods of time, or genetic profiling of hundreds of genes, the need for comprehensive analysis has grown. Both traditional models and neural networks are adaptable and may be better applied when addressing specific requirements of model performance or interpretability of results.

All machine learning paradigms are highly tunable to balance the computational load of the model, time to run, and performance. Typically, multiple model types are applied in a given study because although methods like logistic regression (LR), support vector machines (SVM), and neural networks (NN) are most common, the accuracy performance is often dataset dependent [23]. The models vary in algorithm complexity and transparency, so based on the necessary computational time and sensitivity, model hyperparameters can be tuned to accomplish the desired task. Beyond the classification of samples, models can focus on the selection of the most important features as a way to characterize the variable relations. Feature selection methods can determine any correlation or redundancy when examining a large feature set or improve the features given to an eventual prediction model [24,25]. An overarching goal of any model is the ability to generalize or extend its use to independent datasets. As such, there is a need to ensure a sample size large enough to prevent overfitting, something easy to achieve using high-throughput single-cell measurement systems. 

In this review, we look to cover recent work joining the fields of electrical impedance sensing and machine learning towards the development of more intelligent single-cell diagnostic systems, as shown in Figure 2. To our knowledge, this is one of the first comprehensive looks at machine learning on electrical approaches to improve the standardization and design process for singular cell measurement and analysis. Although machine learning models make decisions based on governing equations, we intend to focus on the applications and would refer the reader to one of many textbooks on the subject [26]. Our discussion includes the more explored method of machine learning as an analytical tool to address common challenges with existing electrical measurement systems. In addition to machine learning as an analytical tool, we cover systems where machine learning is used as an iterative approach to achieve more rapid and cost-effective device development in both microfluidics and sensors, making an argument for more single-cell applications in this design field.

## 2. Machine Learning for Electrical Sensor Data Analysis on Single Cells

Single cell measurements collected using electrical sensors typically fall into the categories of cytometric or spectroscopic. Cytometric measurements reach high sample numbers, however, are limited in the frequency features that can be collected while a cell passes through the measurement gap and the interaction of a cell with a constant electrical field is positionally dependent. Alternatively, spectroscopic measurements collect a larger number of frequency features and properties, however the longer measurement and need for cell trapping limits the number of cell samples. Machine learning is ideal in both cases when compared to traditional statistical methods because of the adaptability to incorporate and compensate for these confounding and limiting factors. In this section, we discuss the ways ML can address the limitations of electrical measurement systems to improve the ability to analytically distinguish between individual cells.

### 2.1. Positional Dependency Compensation

One of the factors most crucial in preventing overfitting and aiding the later generalization of machine learning algorithms is the large sample size necessary. For this reason, electrical impedance measurements in single-cell applications are overwhelmingly conducted using impedance flow cytometry, which is closely related to the previously mentioned optical flow cytometry. However, because the principles of impedance modeling typically rely on the assumption that the cell is subject to a uniform electrical field during measurements, positional changes and size heterogeneity in a cell population can impact accuracy. Considering the small magnitude of most cellular changes in an electrical system, characterizing these confounding factors becomes integral for improving identification of true properties of the cell versus the measurement system. A summary of recent work using machine learning to compensate for the positional dependency of flow cytometry measurements can be found in Table 1.

Considering cytometric measurement limitations, several papers in recent years have worked to establish correction factors to monitor cell location during measurement and improve the classification of particles based on positional compensation. These methods can either rely on the peak amplitude and spacing properties of the time domain cytometric measurement [31] or extracted parameters calculated from the initial measurements, such as opacity [28,30]. For these methods, the accuracy of the model is typically defined as the closeness to the distributive values of the measured parameters. Work from Honrado et al., used a recursive neural network operated in real time to show that based on impedance measurements, particle diameter could be predicted within 0.9 microns, velocity could be predicted within 2.2%, and position could be predicted within 2.4% [31]. Machine learning for this purpose can also assist in monitoring the ability of sheath flow to direct cells to an optimal measurement location [27]. Inclusion of the size, positional, or biomechanical properties of cells has been shown to improve the classification when considered as features for cells of similar types. Apichitsopa et al., generated predictions for similar types of leukemia cells using polarization at three frequencies, size, and deformability with an overall accuracy of 71.4% classification. In this work, they showed that the inclusion of the physical properties alongside the electrical properties improved the accuracy and consistency of the predictions [29]. Based on the discussion presented, a variety of machine learning methods can be used to predict and compensate for the positional dependence of impedimetric flow cytometry readings, making the results of the measurement technique more accurate and reproducible.

### 2.2. Analyzing Dielectric Parameters

One of the goals of multi-frequency electrical measurement is the determination of internal cell properties, most commonly the dielectric properties of the cell membrane and cytoplasm. Determination of these intrinsic properties is possible due to the differential scattering at various frequencies of electrical sinusoidal signals. Based on the work of Foster and Schwan, it is well established that in biological cells and tissues, different compartments dominate the signal at different frequencies [32,33]. In subsequent years, these scattering properties have been further expanded to include specificity of cellular inferences that can be made from each range of scattering [34,35]. Understanding dielectric properties combats one of the main concerns about electrical measurement, which is the difficulty explaining what exactly is causing the measured change within the cell to cause an electrical difference. Dielectric properties can be determined through several methods, two of the most popular of which are dielectrophoresis or impedance analysis. 

Use of the principles of dielectrophoresis (DEP) is a common way to distinguish between cells with different dielectric properties, often without the need for circuit modelling. Dielectrophoresis can determine a unique crossover frequency at which a repulsive or negative DEP signal response changes to an attractive or positive DEP signal response. Without the use of models to differentiate between cell types, work in the DEP field has shown the ability to distinguish stages in colon cancer models [36] and glioblastoma models [37]. Characterizing this unique frequency-based response change in different cell types and in cells under different conditions including after differentiation or drug treatment, has been extensively described in previous work [38]. DEP measurements have also been combined with shell modelling as described in the next section to develop more interpretable results and extract parameters of the nucleus [39]. Although discrimination is possible independent of the dielectric property simulation, machine learning may help enhance our understanding of the correlations between dielectric properties and the physiological properties of different cell types. 

Dielectric properties are determined using impedance measurements through a circuit and shell model, wherein the cell is considered a combination of mixtures which can be polarized with unique properties to define the membrane, cytoplasm, and nucleus. These models are computationally intensive to run, especially in more complex multi-shell models to examine the nucleus and it is often difficult to determine the appropriate parameters for simulation. Despite the complexity of designing and fitting these models, it remains important to expand the understanding of electrical spectroscopic measurements. Without an understanding of dielectric properties, it is difficult to rationalize or justify a choice to shift diagnosis, considering the lack of specificity to a particular intracellular target. Applying an understanding of how electrical properties change with certain diseases makes the attempts to classify cells less of a black box model, where only the inputs and outputs are fully realized. 

In the age of rapid diagnostics and high throughput, there is a need for similarly improved speed in parameter extraction for both dielectrophoretic and impedance models. Neural network models have been used to predict dielectric parameters in real-time for individual cells based on raw impedance values in cytometric systems based on previous simulation fittings [40]. In another work, similar neural network classification strategies have been shown to quickly generate dielectric parameters as a precursor to a rapid classification model to identify cell types. In this example, Tan et al. showed that cytometric constriction channels combined with a feedforward neural network can distinguish different types of similarly size leukocyte cell lines based on four frequency impedance values [41]. In another complex application of the neural network approach, Caselli et al. applied a multi-layer recurrent neural network (RNN) for initial data segmentation followed by a classification scheme using a multiple convolution neural network (CNN) structure to identify red blood cells and nearly identical ghost red blood cells [42]. In this work, impedance measurements at eight frequencies were evaluated to accurately predict cell radius, membrane capacitance, cytoplasm permittivity, and cytoplasm conductivity and classification using these parameters identified the cell types with an accuracy of 96.6%. A comparative summary of these recent works can be found in Table 2.

Alternatively, recent work from Tang et al. uses a maximum length sequence (MLS) system to analyze 512 broadband frequency impedance measurements to calculate the impedance magnitude and phase for each cell [43]. The most easily distinguished range of frequency magnitude and phase were then analyzed using a k-Nearest Neighbor (KNN) learning model to classify adenocarcinoma cells and white blood cells with an accuracy of 98.9%. Based on the models discussed in this section, a variety of learning schemes can be used to (1) improve the real-time identification of cells based on extrapolated dielectric properties from limited frequency information and (2) improve classification between groups based on measured dielectric properties to identify the most relevant frequency regions.

### 2.3. Classification of Cell Differences

Remembering that the ultimate goal of most electrical impedance measurement systems is improving the speed, cost, and overall accessibility of diagnosis, one of the most important challenges to address in a measurement system is the sensitivity to distinguish populations. The applications of this can include identifying healthy from diseased cell states [44,45,46,47], determining the proliferation of patient cells for clinical study [48,49], or quantifying the response of cells to a potential treatment [50,51,52]. In each case, there exist multiple populations representing different changes that can be difficult to determine, especially in cases where cells each have individual responses to treatment or levels of disease. A summary of recent work identifying changes in cellular condition using various machine learning methods for data analysis can be found in Table 3. The benefits of the methods employed in this section are the reduced computational burden and time to prediction saved by model training without circuit fitting.

The effectiveness of classification schemes typically relies on the data type and preprocessing applied as well as the hyperparameters given to the model. The cyclical process of optimizing a model for the data type and the evaluation required to make predictions on new data can be seen schematically in Figure 3. The need for cyclical and thorough evaluation of multiple methods and parameters in a given model is exemplified in work by Jeong et al. wherein they compared the classification accuracy of normal and cancerous cells using a micro-EIS device taking rapid cytometric measurements [44]. The work compared the prediction accuracy of 5 different supervised machine learning schemes as well as a deep learning structure, showing the best accuracy using RF and SVM. In applications identifying the effects of drugs on cells using cytometry, measurements on the same cell type can be difficult to differentiate, requiring processing to both generate appropriate features from the initial signals and determine which features are most effective when given to a classification model. The cyclical nature of these processing steps is readily exemplified in the context of classifying the effectiveness of a treatment on cancer cells from Ahuja et al. [53]. In this work, the change in signal amplitude at four frequencies became features to train an SVM classifier and showed that when compared with traditional live/dead staining using trypan blue, there was impressive correspondence between the two methods. 

There are three overarching machine learning approaches that have been applied to address raw electrical population differences: unsupervised clustering, supervised learning models, and neural networks, among others. The least computationally demanding of these is clustering, an approach that can be either unsupervised for the purposes of visualization or supervised to apply a classification using known data labels. Using a clustering approach, cells become grouped based on proximity to a predicted central position in a feature space. In a similar fashion, support vector machine models generate a decision-making plane in a projected feature space and classify based on where new samples would project to. Many of these classifications are done by artificial neural networks as mentioned earlier, which model the decision-making process of human neurons wherein each node gets multiple inputs and the output is established based on whether the weighted inputs reach an established threshold [57]. ANNs are especially useful for learning hierarchies and tackling more complex non-linear problems or feature relationships [57]. A recent publication distinguished leukocytes using FNN and RNN methods to separate populations using impedance cytometry measurements and demonstrated an increase in classification accuracy to 84.9% and 97.5%, respectively [55]. Considering the highly overlapping properties of these cell types, neural network flexibility and complexity made this subpopulation analysis possible without the typical expensive and time-consuming processing. Artificial neural networks can improve model flexibility and accuracy for complex fitting problems, however they tend to be limited in interpretation, as they are generally approached as a black box model. 

The combined use of feature selection and classification together can provide insights when sensitivity makes the separation of populations difficult. An example of this is our previous work classifying cells in populations using electrical impedance data at 201 frequencies ranging from 9 kHz to 9 GHz to identify changes in nucleus size [58]. In this work, we found that the combination of feature selection using recursive feature elimination (RFE) when combined with SVM both improved the accuracy of predictions but also could identify the most relevant frequency features. The benefit of this is the ability to distinguish the best frequency measurements to explain this highly variable spectra change even in the highly overlapping populations, in which there is an inherent biological variability. In this case, a less computationally demanding model was able to increase the sensitivity of the overall analytical system to internal changes in impedance. 

For work identifying the composition of cells in a solution, clustering or segmentation-based learning methods are the most effective at partitioning the populations. Schütt et al. showed that clustering methods can be used to identify the proportion of myeloblasts compared to the regular blood proportions in samples from patients with acute myeloid leukemia (AML) [48]. This rapid nano-impedance cytometer used impedance measurements and peak analysis to compare the population proportion with results from several optical and electrical techniques including fluorescence-activated cell sorting (FACS) and electrical impedance spectroscopy (EIS), among others. Through both feature selection and classification methods, machine learning can assist in the identification of specific effects of different treatments or classes of cells.

## 3. Machine Learning for Intelligent Design of Microfluidics and Sensors

A unique and emerging application of machine learning is to predict the performance of new measurement paradigms and conditions to streamline the prototyping process. In this way, less time and resources can be spent fabricating and characterizing sensors that may not provide optimal results for the final measurement design [59]. In the process of creating most microfluidic channels or sensing systems in general, there are many steps in the production of a physical system based on a simulated design including characterizing the size, surfaces, and the efficacy of any surface treatment. Machine learning can be applied to the process of design to predict the outcome of certain variable changes without the need to run the physical manifestation through an experiment. The application of this overcomes traditional laboratory limitations in resources and time to develop a successful design. In this section, we organize the history of machine learning driven design processes in the tangential fields of microfluidics and electrical sensing to show the potential for adopting these principles for single-cell problems. While machine learning for design optimization has played a role in these adjacent fields, it remains largely untapped in single cell analysis. Going forward, there may well be a place for the improvement of the design processes to create and produce single-cell focused microfluidics for both manipulation and analysis.

### 3.1. Microfluidics Design and Control

Microfluidic systems are integral for the study of single cells and the development of diagnostic tools that are both rapid and portable. Machine learning can be either applied to the design of these systems or automating the operation of specific fluidic control components [60,61]. Using machine learning in the design of these systems often relies on deep learning and the incorporation of some mathematical model, either based purely on the governing equations of fluid dynamics or software simulation using a program like COMSOL. The incorporation of machine learning is typically a function of reducing the computational load necessary through repetitive simulation of various parameters in the channel with examples typically including the flow rate, channel width, or protruding features. It is also necessary to mention that a key benefit of automating these systems is device translation between research settings so that similar devices and control systems can be created for differing applications [62]. 

Machine learning and microfluidics have been combined in a variety of applications in the field of medicine. Intelligent microfluidic design allows the simultaneous control and analysis of more complex systems, which has been applied mostly in the realm of optical characterization rather than electrical diagnostics. One example is the development of a multiplexing assay to identify Lyme disease using a streamlined process to select relevant antigens on an optically analyzed device [63]. In another instance, an applied assay based on a digital microfluidic sensor was tuned by identifying the features to optimize a particular reaction or yield in each channel. Notably, this was shown using both linear regression and neural networks, showing that either model complexity can characterize and predict the same outputs [64]. Similarly, in the study of bacteria, unique learning-based design systems were used to automate the culture of thousands of microwells to monitor the growth of genetically modified bacteria [65] and monitor the chemotaxis of members in a bacterial community [66]. In addition to these, there have been efforts to incorporate quantitative pharmacology methods into the more efficient design of organ-on-a-chip systems in which systemic effects of circulation and bodily interactions are modeled on a small scale to better predict the complex relationships between chambers [67,68]. The incorporation of various analytical learning methods into the development of microfluidics promises to revolutionize all small-volume sensing applications, however, currently remains understudied in single-cell electrical systems.

Considering that typically, single-cell electrical analysis systems involve the design of microfluidics to isolate the individual cells over a particular sensing region there is a need to incorporate machine learning based methodology to develop more integrated systems. By using smart design choices to create standard practices, every part of the sensing process including control, measurement, and the eventual processing of the impedance measurements from the individual cells can be incorporated.

### 3.2. Electrical Sensor Design and Control

Parallel to the push to incorporate smarter design processes into the realm of microfluidics, electronic sensors are constantly moving to become smaller, faster, and more accessible in the digital world. Inclusion of machine learning to design systems has been emphasized as the critical next step to develop lab-on-a-chip sensors that are sized for easy transport and user friendly enough to move into healthcare environments [69]. The automated and improved design of commercial mechanical sensors has long been posited as the solution to connecting the sensor with the monitored process, enabling more rapid response to system changes and better data retrieval [70]. In this way, the investment using machine learning at the beginning of the sensor production process can reap dividends in its output incorporation within modern smart systems. While this has been shown in biosensor design applications at several levels, the ML-directed design of single-cell sensors has been slower to be adopted. 

Machine learning incorporation into biosensor-based devices has been previously reviewed for the analysis of biological molecules and tissues in several publications [71,72,73]. Recent highlights in the incorporation of machine learning designed devices includes the work from Govindaraju et al., which identifies white blood cell count on a smartphone integrated system for ease of measurement display [74]. Alternatively, machine learning was used to both design the monitoring system and development of tissue growth on a bioscaffold using electrical impedance spectroscopy by Shohan et al. [75]. The design of this system was critical in its ability to not impact the tissue health, making it a viable option for the analysis of patient cultures for future graft or transplant applications. By saving time and resources in the production of clinical devices, there is more room to adapt to developing clinical needs during the process of translation.

While point-of-care (POC) devices remain one major rationale behind single-cell electrical sensing, the field mostly remains at the research phase. Newer generations of diagnostic devices result in the production of more information and necessitates more stringent standards of accuracy, safety, and understanding as automation becomes incorporated. The combination of device and computational systems allows scientists to actively parse this information, however practical device design and control becomes critical for them to translate from the benchtop to the doctor’s office. Reyes et al., explains the need for practical standards in microfluidics to bridge this gap and also increase the accessibility for diagnostic devices that anticipate non-expert users [76]. The improvement of the design that machine learning could create for single-cell measurement and analytics is essential for the standardization that would be necessary to create any commercialized medical system.

A primary example of machine learning in the control of microfluidics and electrical sensors for single-cell cytometry is recent work from Wang et al. [77]. In this work, real-time analysis of cytometer electrodes using a convolution neural network provided feedback-based controls to the pumps running the sample. Based on the real-time cytometric measurements on the cells, the system automatically adjusted the flow rate to control the number of samples measured with an accuracy of greater than 90%. This presents an interesting and useful construct for the optimization of measurement patterns and the prevention of clogging within microfluidic systems. The success of these efforts belies the benefits of fully integrated designs in all aspects of single-cell measurement systems.

## 4. Machine Learning Analysis of Non-Electrical Single-Cell Measurements

In the past few years, single-cell characterization methods have experienced pushes to incorporate machine learning for data analysis, most notably in the fields of Raman spectroscopy, optical flow cytometry, and genomic profiling. The success in these similar single-cell processing fields provides an aspirational framework for the adaptation of standardized data processing and machine learning methods in the electrical field. While not yet perfected in any field, the widespread use and greater historical context experienced in these other single-cell measurement types shows the advantage of comparable data to enhance the study of wider populations.

Raman spectroscopy is a method that uses the vibrational properties of a material to generate a spectrum that describes the chemical composition of the cell [78]. Machine learning has been combined with this data type for the purpose of classifying differing cell types, both mammalian [79,80] and bacterial [81]. Optical flow cytometry is a method that relies on images of rapidly moving cells, typically characterized by either deformability, size, or intensity of a targeted fluorescent label. Several reviews have covered the combination of this method with machine learning to automate the detection of specific subpopulations [82,83], improve high-speed analytical throughput [84], and address the accessibility of cancer diagnosis in clinician-limited settings [85]. Many algorithms and applications have been developed to address the analysis of single cells in these non-electrical fields while the measurement technology has struggled to become more cost-effective and higher throughput. This directly opposed the concerns seen in the field of electrical measurement where the devices are already developing the throughput and cost-effectiveness to address the data needs, but there is a distinct need for standard algorithms and analysis methods.

Genomic profiling in the ‘-omics’ field can incorporate analysis of genomic, transcriptomic, proteomic, or epigenomic data to track the changes in both genetic content and expression in singular cells belonging to the same population. A vast array of papers on this topic have been published in recent years, there have also been several reviews to summarize the work in this field [86,87,88]. Genetic analysis of individual cells using RNA has been used to map associated changes within many cell types affected by acute myeloid leukemia [89] and analyze the quality of laboratory derived macrophage treatments [90]. Incorporation of methods such as t-distributed stochastic neighbor embedding (t-SNE) or partition-based graph abstraction (PAGA), as seen in Figure 4, for visualization has also enabled the comparison of properties shared among cell types across all stages of development from fetal stem cells to differentiated adult cells using genetic information [91]. 

As previously mentioned, there are many standard algorithms for ‘-omic’ analysis of single-cell data which have been established and published online, making this single-cell method one of the most accessible after obtaining the expensive sequencing equipment. The success of RNAseq and similar algorithms in the field of genetic sequencing shows the ability of a field to adopt, standardize and communicate these more complex data analysis methods and points the direction the electrical single-cell analysis field can aspire to.

## 5. Conclusions and Future Outlooks

Many challenges exist when meeting the criteria required for adopting single cell analysis to new applications both in research and clinical settings. Primarily, the need for a sufficient number and distribution of samples to capture the true properties of the entire population. Additionally, there is a need to ensure reproducible measurements from the systems that can be used to consistently train and validate the machine learning models. The collection of repositories of consistent and sizeable data structures is a critical next step to generating new integrated methods of design and analysis to compare the complex property changes in single cells biophysically, genetically, and metabolically. Growth in this field and the compilation of larger datasets could enable an electrical profiling capability on par with the development of the human genome project, however generating a data type easier to collect and analyze in a point-of-care setting. 

Future research in the combination of machine learning paradigms with electrical single-cell sensing can leverage the design principles and processes to branch into wider applications. Most research around this population analysis of single cells is centered around flow cytometry, due to the large sample number that can be collected and the established measurement processes and equipment. However, as discussed in the design section, iterative design prediction could be used to create more rapid spectroscopic measurement systems wherein larger numbers of frequencies or smaller footprint devices can improve the data quantity or accessibility of diagnostic tools. In addition, real-time classification of samples, especially in blood testing, could be incredibly useful in a clinical setting to decrease the time from processing to diagnostic results. Because machine learning can produce rapid and accurate classification of individual cells, it would be incredibly useful for identifying circulating tumor cells in blood samples or identifying alterations in blood cell properties to indicate disease for screening. The future is especially bright considering the incorporation into interpretable AI to address the black-box model concerns and improve the accessibility of machine learning models for the general public. 

A main challenge that integrated and standardized practices can also help address is the individualized nature of the performance of different machine learning categories with each dataset. The performance depends highly on the features measured themselves, the complexity of the relationship between the variable features, and the amount of computational power required to address the classification challenge. As shown in Table 4 below, each method does have associated pros and cons, making different paradigms ideal for different problems and types of impedance information collected or fitted. Methods that handle deeper complexity of relationships are typically more computationally demanding and less interpretable. These are more generalized evaluations, and the performance is generally dependent on the data itself, making a wide-sweeping, thoughtful, and eventually standardized approach uniquely beneficial for future efforts in this field. 

Electrical single-cell sensing is one of the most viable options for accessible diagnostic systems, especially in resource limited settings where permanent infrastructure or trained personnel may be limited. Machine learning enables the incorporation of analysis into more inclusive, small footprint devices and systems that make it easy to take a rapid and accurate tool for diagnosis anywhere in the world. While the incorporation of these analysis methods has revolutionized the information handling in traditional electrical sensing fields, there remains the potential to revise devices and measurement schemes based on machine learning suggestions in the design process. This could include iteratively determining the frequencies of interest and adjusting measurement design accordingly or automating control systems in a way that reacts to common problems in microfluidic systems like clogging or balancing throughput with measurement quality. By learning from the applications already supplied in general microfluidic or assay design, the field of single-cell electronics has the potential to move into smaller, inclusive, and accurate tools for diagnosis, using intelligence to overcome the posed challenges.

## Figures and Tables

**Figure 1 sensors-23-05990-f001:**
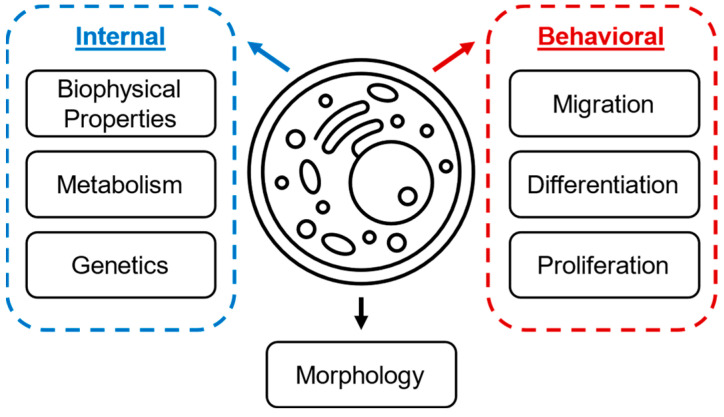
Summary of individual cell properties that can be measured to distinguish populations.

**Figure 2 sensors-23-05990-f002:**
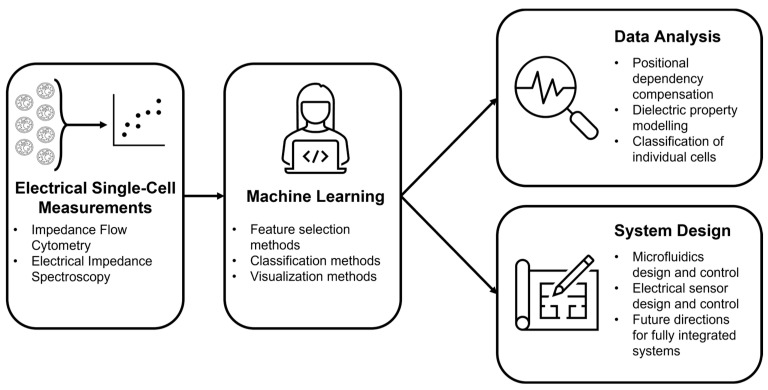
Structural overview of topics covered over the course of this review.

**Figure 3 sensors-23-05990-f003:**
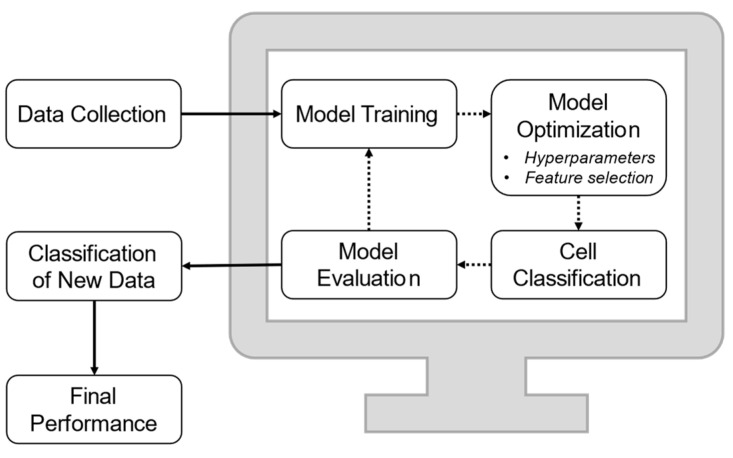
Schematic demonstrating basic processing in ML classification model training and to determine final performance on a population of treated cells.

**Figure 4 sensors-23-05990-f004:**
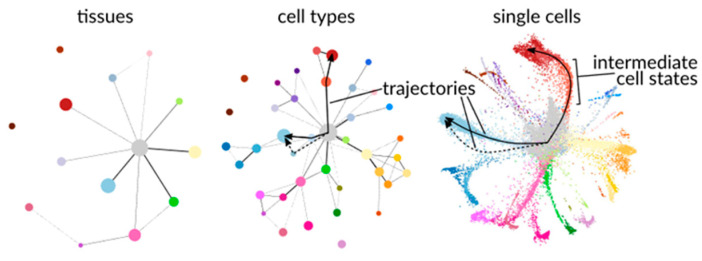
Visualization using partition-based graph abstraction (PAGA) of different levels of cell property clustering during the process of differentiation. Reprinted with permission from Ref. [91]. 2020. *Genome Biology*.

**Table 1 sensors-23-05990-t001:** Summary of recent publications using various ML methods to compensate for size and positional dependency of flow cytometry measurements on single cells.

Learning Category	ML Method	Achieved Accuracy	Application	Citation
Deep Learning	NARX NN	4.3 × 10^−5^ Normalized Mean Square Error	Predicting Particle Impedance and Location in Sheath	[27]
Supervised	Linear Regression	37% improvement in size distinction	Positional Dependency Compensation and Size	[28]
Supervised	Random Forest	71.4% using size, deformability, and polarization	Using position and size in addition to electrical measurements to enhance classification	[29]
Supervised	Linear Regression	Accuracy within 1.5 µm of the height	Positional and size determination using opacity and impedance	[30]
Deep Learning	RNN	Within 0.09 µm for diameter, 2.2% for velocity, 2.4% for position	Predicting cell X and Y position based on properties of time domain curve	[31]

**Table 2 sensors-23-05990-t002:** Summary of recent work using various ML methods to predict dielectric parameters of mammalian cells.

Learning Category	ML Method	Achieved Accuracy	Application	Citation
Shallow Learning	FCN	94.6%	Predicting dielectric parameters in real-time to identify cell type	[40]
Shallow Learning	Feedforward NN	90.5%	Determining dielectric parameters in constriction microchannel and identifying cell type	[41]
Deep Learning	RNN, CNN	96.6%	Predicting dielectric parameters in real time for classification	[42]
Unsupervised	KNN	98.9%	Using Extracted Dielectric Parameters to train classification model	[43]

**Table 3 sensors-23-05990-t003:** Summarized recent works applying various ML methods to identify cellular responses to disease or treatment.

Data Type	Learning Category	ML Method	Achieved Accuracy	Application	Citation
Impedance Cytometry	Supervised	SVM	95.9%	Identifying the efficacy of drug treatment on cancer cells	[53]
Impedance Cytometry	Unsupervised	KNN	98.4%	Identify drug treatment efficacy using electrical and optical flow cytometry data	[50]
Impedance Cytometry	Supervised, Deep Learning	LR, KNN, DT, SVM, RF, BPNN	91.7% using RF and SVM	Distinguish cancerous and healthy bladder cells	[44]
Electrical Impedance Spectroscopy	Supervised	QDA, SVM, Ensemble Bagged Tree	99.5% using Ensemble Tree	Detecting Surface Protein in Severe Endometriosis	[45]
Electrical Impedance Spectroscopy	Shallow Learning	LSTM RNN	91%	Identifying proliferating and differentiated patient cells	[49]
Electrical Impedance Spectroscopy	Supervised, Shallow Learning	MLE, LDA, BPNN	100%	Identifying strains of gram-negative bacteria that commonly contaminate food	[46]
Impedance Cytometry	Supervised	SVM	9.2% Detection Error	Identification of antibiotic-susceptible bacteria in real time	[51]
Impedance Cytometry	Shallow Learning	BPNN	98%	Identify MCF-7 cell with treatments based on electrical and biophysical properties	[52]
Impedance Cytometry	Unsupervised	Clustering	1–3% Deviation from True Proportions	Identifying proportion of blood cells in AML patients and healthy controls	[48]
Impedance Cytometry	Supervised	Gaussian SVM	99.8%	Identify CTC from WBC in focused serpentine channel	[47]
Impedance Cytometry	Supervised	LDA, SVM	91.2%	Distinction of PBMC’s in mixed solution and cancer cell lines	[54]
Impedance Cytometry	Shallow Learning	FNN, RNN	84.9% and 91.2%	Identifying subpopulations of leukocytes	[55]
Impedance Cytometry	Supervised	KNN, SVM, LR, DT, AdaBoost, Gaussian Naïve Bayes	92.5%, 93.7%, 90.2%, 88.5%, 90.6%, 84.3%	Classifying response of prostate tumor and cancer associated fibroblasts to treatment	[56]

**Table 4 sensors-23-05990-t004:** Summarized pros and cons of mentioned machine learning classification methods.

ML Method	Pros	Cons
Support Vector Machine	Complexity Interpretability	Computational Demand
Neural Networks	Complexity	Computational Demand Interpretability
K-Nearest Neighbor Clustering	Interpretability Computational Demand	Complexity
Decision Trees	Interpretability Computational Demand	Complexity
Random Forest	Complexity Interpretability	Computational Demand
Logistic Regression	Interpretability Computational Demand	Complexity

## Data Availability

Data sharing not applicable—no new data generated.

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
