# Peer review of "Recent Approaches to Design and Analysis of Electrical Impedance Systems for Single Cells Using Machine Learning"

_sensors, 2023, doi:10.3390/s23135990_

Round 1

Reviewer 1 Report

Authors reviewed the employment of ML in the electrical impedance measurement for single cells. I have concerns about this work as follows.

1. There is no figure to help explain what the authors would like to tell. It is quite difficult for a reader who is new in this field. The review article should support a new reader to extract the information easily.

2. The key messages of this work are in Section 3. However, no well-organized and adequate information is provided.

3. In Section 4, authors add another technique (non-electrical) that should be removed. It reflected that the knowledge about ML vs impedance measurement for single cells is not established yet. This can be improved. For example, authors should add information about the design of system that is suitable for a specific ML (or vice versa).

4. If it is possible, authors should discuss about pro and con of each ML technique for impedance measurement of single cells.

Author Response

Thank you for your comments, please see the attachment. 

Reviewer 2 Report

The paper is well written, but some points should be corrected before publishing the paper:

1- What is the code used in the graphs 

2- The English are good, but there are some typos, There are many grammatical, typographical errors in the paper. They can be easily seen from the text. Read carefully.

3- The Novelety and the main contribution should be cleared

4-The conclusion must be rewritten.

5-It is compulsory to mention the simulation software and give the codes or application (which is made by authors) as an appendix

6-Regarding Table 1, some new learning categories should be added 

7-Which one was the best categorie

English are good

Author Response

(The authors gave the same response as above.)

Reviewer 3 Report

The authours quite well reviewed recent studies of electrical impedance systems for single cells using ML. However, I am just wondering whther the section 4 is necessary for this review paper. Also, it would be better if authors could do minor english corrections before the final publication.

I recommend authors check English in this review manuscript one more time. For example, the sentence on page 4 line 176 sounds little odd.

Author Response

(The authors gave the same response as above.)

Reviewer 4 Report

The paper reports a review about incorporation of machine learning (ML) to improve the field of electrical single cell analysis by addressing the design challenges to manipulate single cells and sophisticated analysis of electrical properties that distinguish cellular changes. The paper highlight the opportunity to build on integrated system technologies to address common challenges in data quality and generalizability to save time and resources at every step in electrical measurement of single cells.

Overall the paper is well organized. The paper is well written and presented while its theoretical foundation has an interesting novelty. The present work’ connection with the open literature background is well and adequately justified.

I my opinion some specific examples should be presented of how are used these methods presented in the paper. The authors should explain some governing equations of the methods reviewed.

At page 9, row 334, you should correct the end of paragraph: “ …..relationship54.”.

The manuscript should be revised before further consideration of publication.

Author Response

(The authors gave the same response as above.)

Round 2

Reviewer 1 Report

The authors have improved the quality of the work by clarifying the mentioned issues very well. The revised work is acceptable for publication.

Reviewer 2 Report

The paper can be accepted for publication